# The Effect of Low and High Dose Deoxynivalenol on Intestinal Morphology, Distribution, and Expression of Inflammatory Cytokines of Weaning Rabbits

**DOI:** 10.3390/toxins11080473

**Published:** 2019-08-13

**Authors:** Wanying Yang, Libo Huang, Pengwei Wang, Zhichao Wu, Fuchang Li, Chunyang Wang

**Affiliations:** Shandong Provincial Key Laboratory of Animal Biotechnology and Disease Control and Prevention, Shandong Agricultural University, 61 Daizong Street, Taian City 271018, China

**Keywords:** deoxynivalenol, intestinal morphology, inflammatory cytokines, weaning rabbit

## Abstract

Deoxynivalenol (DON) is a potential pathogenic factor to humans and animals, and intestinal tract is the primary target organ of DON. Data concerning the effects of DON on rabbits are scarce, especially for weaning rabbits. In this study, 45 weaning rabbits (35 d) were randomly and equally assigned into three groups. Group A was fed basic diet, while groups B and C were added DON at 0.5 mg/kg BW/d and 1.5 mg/kg BW/d, respectively, based on the basic diet. The experiment lasted for 24 days and the intestinal morphology, expression, and distribution of several cytokines in intestinal segments have been examined. The results indicated that ADG decreased while F/G increased significantly compared with the control group after DON added at 1.5 mg/kg BW/d. Some of the morphometric parameters (villi length, crypt depth, and goblet cells density) changed after DON was added. Meanwhile, the concentration as well as the expression levels of relative protein and mRNA of IL-1β, IL-2, IL-6, and IL-8 increased significantly. The immunohistochemistry results illustrated that the quantity and distribution of positive cells of inflammatory cytokines were changed after DON was added. In conclusion, the addition of DON damaged the intestinal morphology and changed the distribution and expression of inflammatory cytokines. The toxic effect depended on the dosage of DON.

## 1. Introduction

Deoxynivalenol (DON), also known as vomitoxin, is one of the most commonly detected trichothecene mycotoxins. Because of the high detection rate in food and feed worldwide, DON has become a potential pathogenic factor to human and animal species [1,2]. Intake of DON usually leads to various toxic effects, such as diarrhea, emesis, destruction of intestinal mucosa, low body weight gain, and damage in immune function [3,4,5].

DON is mainly absorbed by the intestinal tract, which served as the first target organ of mycotoxins attack, inducing intestinal lesions, regulating intestinal immune response, changing intestinal immune barrier function, and causing intestinal inflammation [6]. Destruction of the integrity of the mechanical barrier of intestinal tract and induction of intestinal inflammation may be one of the main ways for DON to exert toxicity [7,8]. Some research concluded that the intensity of toxic effect of DON depends on dose, exposure time, toxin purity, and route of attack [9,10,11,12].

Interleukin, as an important component of cytokines, plays a crucial role in cell–cell interaction, immune regulation, and inflammation [13]. A variety of immunomodulatory effects were induced by DON, which may be related to the increased susceptibility to intestinal inflammatory diseases [14,15,16]. The phenomenon that DON induces produce immune stimulation or immunosuppression, and the reaction is related to its dose, frequency, and exposure time. Low concentration of DON can promote rapid mucosal inflammatory response and generate risks of induced chronic intestinal inflammation, such as inflammatory bowel disease, while high dose of DON inhibits immune response [17]. Many research studies proved that the DON affects the cytokines and immune function. However, there are few data on the relationship between the distribution and expression of cytokines of DON on different intestinal segments.

The research results on the effect of DON on intestinal tract injury in different animals are inconsistent, which may be due to different sensitivity of animals to DON, and the different absorption, distribution, metabolism, and elimination of DON by animals themselves [5,7]. Rabbit meat is an important part of the human diet. The complete rabbit feed is composed of various raw materials, so the pollution of DON in the feed material poses potential harm to animals and human health [18,19,20]. However, the researches concerning DON toxicity are mainly concentrated on swine and poultry, whose intestinal structure, flora distribution, and physiological functions are different from those of rabbits [21]. Surprisingly, the research concerning the effect of DON on rabbits was limited, especially on weaning rabbit. Therefore, this study was designed to evaluate intestinal histomorphology and the expression and distribution of related proinflammatory factors in weaning rabbits induced by DON at higher and lower doses.

## 2. Results

### 2.1. Production Performance of Weaning Rabbits

In this research, the average daily gain (ADG) was decreased significantly (*p* < 0.05), while the feed/gain ratio (F/G) increased significantly (*p* < 0.05) compared with the control group and lower-dose group after DON added at 1.5 mg/kg BW/d. The results showed that the average daily feed intake (ADFI) had no significant difference (*p* > 0.05) among the three groups (Table 1).

The relative weight of liver and spleen of weaning rabbits decreased significantly (*p* < 0.05) in groups B and C in which DON was added at lower and higher dose, respectively, compared with control group. Meanwhile, the relative weight of kidney and stomach showed no significant difference (*p* > 0.05) among the groups (Table 2).

### 2.2. Biochemistry Factors in Serum of Weaning Rabbits

The concentrations of ALT, ALP, CK, and LDH increased significantly (*p* < 0.05) in the groups where DON was added at lower and higher doses compared with the control group, while that of AST, ALB, and UA increased significantly (*p* < 0.05) only in the group where DON was added at a higher dose. The concentrations of TP, UREA, GLU, and TG in serum had no significant difference (*p* > 0.05) among the groups (Figure 1).

### 2.3. The Immunoglobulin Concentration in Serum of Weaning Rabbits

The concentration of IgM reduced significantly (*p* < 0.05) in groups B and C where DON was added at lower and higher doses, respectively, compared with the control, while the concentration of IgG and IgD lessened significantly (*p* < 0.01) only when DON was added at a higher dose. The concentrations of IgA had no significant difference (*p* > 0.05) among the groups (Figure 2).

### 2.4. Intestinal Histomorphology of Weaning Rabbits

The villi heights (VH) were reduced significantly (*p* < 0.05), as well as the depth of crypt (CD) enhanced significantly (*p* < 0.05) in three intestinal segments in groups B and C compared with the control group. By comparison, the downward trend of VH in duodenum and the upward trend of CD in duodenum were most obvious among three intestinal segments, and this trend is related to the dosage of DON (Figure 3a).

The representative morphologies in different groups are illustrated in Appendix A. It can be observed that the mucosa of duodenum, jejunum, and ileum were all injured and presented histological lesions in the rabbits of groups B and C where DON was added at lower and higher doses, respectively, especially for ileum at higher dose. In addition, the epithelial cells of the intestinal villi were destructed severely. The degree of injury was closely related to the dosage of DON, that is, the higher the dose of DON, the more obvious toxic effects were observed.

The number of goblet cells in three intestinal segments increased significantly (*p* < 0.05) in groups where DON was added at lower and higher doses compared with the control. By comparison, the upward trend of goblet cells in ileum was most obvious among the three intestinal segments. Some slice photographs are illustrated in Appendix A.

### 2.5. The Expression and Distribution of Inflammatory Factors

#### 2.5.1. IL-1β

The content of IL-1β in jejunum and ileum were enhanced significantly (*p* < 0.05) in groups where DON was added at lower and higher doses compared with the control, while that of IL-1β in duodenum increased significantly (*p* < 0.05) only in the higher-dose group (Figure 4a). The relative mRNA expression of IL-1β in jejunum increased significantly (*p* < 0.05) in the higher-dose group compared with the control group. The upward trend of relative mRNA expression can be observed in duodenum and jejunum after DON was added; however, there was no significant difference (*p* > 0.05) among the groups (Figure 4b). The relative protein expression of IL-1β in jejunum increased significantly (*p* < 0.05) in groups where DON was added at lower and higher doses compared with the control group, while that of IL-1β in duodenum and ileum enhanced significantly (*p* < 0.05) only in the higher-dose group (Figure 4c).

The IOD and SIOD value of positive cells for IL-1β in three intestinal segments increased significantly (*p* < 0.05) compared with the control group after DON was added, especially at higher-dose group (Figure 4d). By comparison, the most significant changes were exerted in jejunum compared among the three intestinal segments. The representative photography of immunohistochemistry (IHC) in different intestinal segments are illustrated in Appendix A. The results displayed that the positive reactants of IL-1β were distributed in the lamina propria around intestinal gland principally in the control group. The positive reactants of IL-1β in intestine segment were mainly distributed in the lamina propria around intestinal gland but less in villi epithelial cells after feeding the rabbits with DON.

#### 2.5.2. IL-2

The concentration of IL-2 in three intestinal segments enhanced significantly (*p* < 0.05) only in the group where DON was added at higher dose compared with the control group, while there was no significant difference (*p* > 0.05) between groups A and B (Figure 5a). The relative mRNA expression of IL-2 in duodenum and jejunum increased significantly (*p* < 0.05) only in the group where DON was added at higher dose compared with the control group. However, there was no significant difference in ileum among the groups (Figure 5b). The relative protein expression of IL-2 in three intestinal segments increased significantly (*p* < 0.05) only in the group where DON was added at higher dose compared with the control group. The upward trend of IL-2 in duodenum was most obvious among the three intestinal segments (Figure 5c).

The IOD and SIOD values of positive cells for IL-2 in three intestinal segments increased significantly (*p* < 0.05) in groups where DON was added, especially at the higher-dose group, compared with the control group (Figure 5d). It can be seen from Appendix A that the positive reactants of IL-2 were mainly distributed in the intestinal gland epithelial cell but less in the lamina propria of villi in control group. However, the positive reactants of IL-2 were distributed not only in intestinal gland epithelial cell but also in the lamina propria of villi compared with the control group after DON was added.

#### 2.5.3. IL-6

The concentration of IL-6 in duodenum and jejunum increased significantly (*p* < 0.05) in groups where DON was added at lower and higher doses compared with the control group, while that of IL-6 in ileum increased significantly (*p* < 0.05) in the higher-dose group (Figure 6a). The relative mRNA expression of IL-6 in duodenum and jejunum were higher significantly (*p* < 0.05) in groups where DON was added at lower and higher doses compared with the control group. There was no significant difference in ileum among the groups (Figure 6b). The upward trend was observed in three intestinal segments among all groups. However, the relative protein expression for IL-6 in jejunum enhanced significantly (*p* < 0.05) in the higher-dose group compared with the control group (Figure 6c).

The IOD and SIOD values of positive cells for IL-6 in three intestinal segments increased significantly (*p* < 0.05) in groups where DON was added, especially in the higher-dose group, compared with the control (Figure 6d). It can be seen from Appendix A that the positive reactants of IL-6 were mainly distributed in the lamina propria of intestinal gland in control group. However, the positive reactants of IL-6 were distributed in intestinal gland, epithelial cell of villi and the lamina propria of villi in DON-added groups compared with the control group.

#### 2.5.4. IL-8

The concentration of IL-8 in jejunum and ileum increased significantly (*p* < 0.05) in groups where DON was added at lower and higher doses compared with the control group, while that of IL-8 in duodenum increased significantly (*p* < 0.05) only in the high-dose group (Figure 7a). The relative mRNA expression of IL-8 in duodenum was higher significantly (*p* < 0.05) in groups where DON was added at lower and higher doses compared with the control group, while that of IL-8 in jejunum increased significantly (*p* < 0.05) only in the high-dose group (Figure 7b). There was no significant difference in ileum among the groups. The same trend was observed in the relative protein expression of IL-8 (Figure 7c).

The IOD and SIOD values of positive cells for IL-8 in the three intestinal segments enhanced significantly (*p* < 0.05) in groups where DON was added, especially in the higher-dose group, compared with the control group (Figure 7d). The most significant changes were exerted in duodenum among the three intestinal segments. It can be seen from Appendix A that the positive reactants of IL-8 were mainly distributed in the lamina propria of intestinal gland in the control group. However, the positive reactants of IL-8 were distributed in intestinal gland, epithelial cell of villi, and the lamina propria of villi in DON-added groups compared with the control group.

## 3. Discussion

### 3.1. The Toxicity of DON and Animal Species

DON is a kind of mycotoxin, originating from *Fusarium* species, usually monitored in cereal commodities. Many animal species are sensitive to DON and the sensitivity from high to low was pigs, mice, rats, poultry, and ruminants [22]. The representative clinical symptoms of acute DON poisoning include abdominal pain, diarrhea, vomiting, and anorexia [23]. Earlier studies have revealed that the minimum emetic dose of DON in pigs was 100 μg/kg BW [24], and that of DON in cats and ducklings were 400 μg/kg BW and 10 mg/kg BW, respectively [25,26]. The minimum dose for rodents to cause an observable effect was 0.1–0.15 mg/kg BW/d, whereas, this minimum dose for swine was 0.03–0.12 mg/kg BW/d [27].

The most common effects of intake of DON-contaminated diet on animal species for a long term are decreased production performance and nutritional efficiency, which are significantly different from species to species. Earlier studies indicated that DON at 0.04–0.08 mg/kg BW for pigs reduced feed intake, whereas DON at 0.48 mg/kg BW can lead to complete rejection [27]. Most monogastric species are extremely sensitive to DON at chronic and sub-chronic exposure, which can result in inhibition of growth and weight gain. However, there was no decline in production and no signs of illness after cows were fed with DON at 6.6 mg/kg BW for five days or at 0.6 mg/kg BW for six weeks. It is generally considered that the different sensitivity to DON among animal species is not only related to the different metabolism, absorption, and distribution of DON, but also related to the different distribution and structure of intestinal flora.

The effects of DON on rabbits are rarely observed. Limited data showed that the intestinal propulsion was reduced after the rabbits were fed at 1 mg/kg BW [28]. Another research proved that the production performance reduced significantly after pregnant rabbits were fed with DON at 0.3 and 0.6 mg/kg BW, but teratogenic effects were not detected [20]. The production performance of rabbits have not been affected by the addition of DON at 10 mg/kg feed, but the immune response has been modulated [29]. In this study, two doses of DON (0.5 mg/kg BW and 1.5 mg/kg BW) were fed to weaning rabbits for 24 days, and the lower feed efficiency and weight loss were detected only at 1.5 mg/kg BW. The clinical symptoms such as diarrhea, vomiting, and death, however, have not been observed even at higher dose. In conclusion, rabbit’s sensitivity to DON is lower than pig’s but higher than chicken’s from the perspective of clinical symptoms and weight loss, and this may be due to the structure and microbial flora in intestine of rabbits.

### 3.2. DON and Intestinal Morphology

Intestinal tract is a critical defense barrier for human and animals [30]. Addition of 2–3 mg/kg DON-contaminated feed in pigs injured gastric and intestinal epithelial cells and led to the inflammatory response [31,32]. Long-term consumption of the contaminated diets resulted in morphological and histological lesions [8,33]. Intestinal barrier of broiler chickens was disrupted by DON through destroying the absorption, permeability, and other intestinal functions, and further led to lower production efficiency and health damage in animals [34].

The addition of DON can lead to several histological changes at moderate to severe level, such as apical necrosis and villous atrophy and fusion. Meanwhile, DON can reduce the villi height, crypt depth, and goblet cell density significantly [35]. This study analyzed the morphological integrity in intestinal tissue of weaning rabbits induced by DON. The results showed intestinal villi were not integrated and several histological parameters (small intestinal glands, villi height, and crypt depth) were damaged severely after feeding the rabbits with DON. The degree of injury was closely related to the dosage of DON. The most significant injury was exerted in ileum compared among three intestinal segments. On the whole, the DON toxicity is closely related to the integrity of intestinal morphology.

The goblet cells not only produce mucoprotein, but also secrete some factors which can make epithelial renewed and repaired [36]. However, the study on the goblet cells induced by DON was less studied. Limited results indicated that the number of goblet cells in villi and crypts decreased significantly after DON was added in vitro and in vivo [36,37]. The results from this study proved that the number of goblet cells increased significantly, especially at the higher dose.

### 3.3. DON and Inflammatory Factors

The immune system is very sensitive for DON, which could stimulate or suppress the immune system according to the dosage, frequency of exposure, and timing of DON addition [38]. Currently, the researches on DON-induced immunotoxicity mainly focused on pig, chicken, rodent, and some cell lines [8,9,13]. The intake of DON-contaminated feed can increase the susceptibility to infections significantly, leading to reactivation of chronic infection, and subsequently reducing the vaccine efficacy [39]. In this study, with the increase of DON dose, the expression and concentration of IL-1β, IL-2, IL-6, and IL-8 displayed an upward trend. The up-regulation of IL-1β, IL-2, and IL-6 in jejunum but that of IL-8 in duodenum was the most significant among the three intestinal segments.

Many researchers proved that the DON affects the cytokines and immune function [40,41]. A few research studies, however, focused on the distribution of inflammatory factors in intestinal segments induced by DON. Therefore, we further studied the distribution of some inflammatory factors in different intestinal segments caused by DON by IHC method. The results proved that the distribution of positive reactants of IL-1β, IL-2, IL-6, and IL-8 in intestinal segments were different from the normal control group. Furthermore, the distribution of positive reactants of these inflammatory cytokines had been changed after the rabbits were fed with DON. The number of positive reactants of IL-1β, IL-2, IL-6, and IL-8 in duodenum, jejunum, and ileum were increased significantly compared with control group after DON was added, especially at higher-dose group. The most significant changes of IL-1β, IL-2, and IL-6 were exerted in jejunum, while that of IL-8 was observed in duodenum compared among three intestinal segments.

Previous studies have proved that DON can inhibit the secretion of several immunoglobulins [42,43,44]. Our results showed that DON has inhibitory effect on immunoglobulin, especially at the high concentration. This result proves that DON can not only activate intestinal immune system, but also inhibit its immune activity. The intestinal tract of rabbits is different from domestic animals such as pigs and poultry. Thus, we should not only continue to study the mechanism of DON on rabbits in future research, but also look for some biological agents that degrade toxins and are suitable for rabbits to eat in order to protect animal health and improve food safety.

## 4. Conclusions

In conclusion, the addition of DON induced the lower feed efficiency and weight loss compared with the control group, as well as damaged the integrity of intestine segment, changed the distribution and expression of inflammatory cytokines. The toxic effect depended on the dosage of DON.

## 5. Materials and Methods

### 5.1. Ethics Statement

All animal experiments complied with the regulations on the ethical use of experimental animals issued by the Ministry of Science and Technology (Beijing, China). The protocol was permitted by the Animal protection committee of Shandong Agricultural University (ACSA-2018-032).

### 5.2. Diets, Animals, and Sampling

The ingredients of the basic diet, which was based on de Blas and Wiseman (1998), are shown in Table 3. The content of mycotoxins, including DON, zearalenone (ZEA), and aflatoxin B1 (AFB1), were analyzed by LC-MS/MS method. DON standards (C15H20O6, purity >98%) were purchased from Triplebond Company (Guelph, Canada).

Forty-five healthy weaning Rex Rabbits (35-day) were equally and randomly assigned into three groups with 15 rabbits per group (879 ± 17.62 g in mass). Group A was the control group which fed rabbits with basic diet, while groups B and C were treated with basic diet adding DON at lower dose (0.5 mg/kg BW) or at higher dose (1.5 mg/kg BW) in water every day, respectively. The method of DON addition as follows: standard DON was added into sterile distilled water to prepare two kinds of DON mother liquor (1.5 mg/mL and 0.5 mg/mL) for later use. The calculated DON mother liquor, then, was injected into the water fountain containing 20 mL water, which was used to feed weaning rabbits to ensure the correct DON intake every day. After the 20 mL of DON-added water had been consumed, ordinary water was added into the water fountain, and the rabbits were allowed to drink water freely.

The weaning rabbits were placed in metabolism cages (60 × 40 × 40 cm) individually. The rabbits were maintained in a closed building with a semi-controlled environment, and the room temperature was kept between 18–28 °C. The experiment lasted for 31 days, including 7 days of adaptation and 24 days of experiment. The residual feed was collected every day. After the initial and final weights of weaning rabbits were obtained, the value of ADG (average daily gain), ADFI (average daily feed intake), and F/G (feed/gain ratio) were calculated according to the conventional method.

On day 24, 30 rabbits (10 rabbits from each group) were selected for blood collection via ear edge vein at 8:00 am. The serum samples were separated by centrifuging their blood samples at 1500 g for 10 min, then stored at −20 °C for further analysis. Five rabbits (empty stomach) were selected randomly in each group and euthanized using electric shock. After the liver, kidney, spleen, and stomach were removed and weighed, the relative organ weight (g/kg) was calculated. The samples of duodenum, jejunum, and ileum in middle section (about 10 cm in length) from each group were collected. A part of intestinal samples were fixed with Bouin’s solution after washing by 0.9% NaCl solution, while the rest were stored at −70 °C immediately for further experiments.

### 5.3. Biochemical Analysis

Serum biochemical indicators, including TP (total protein), CK (creatine phosphokinase), ALT (alanine aminotransferase), ALB (albumin), GLU (glucose), UA (Uric Acid), TG (triglyceride), ALP (alkaline phosphatase), AST (glutamic oxalacetic transaminase), UREA (blood urea), and LDH (lactic dehydrogenase), were analyzed through the Biochemical Analysis Center in central hospital in Tai’an, Shandong, China.

### 5.4. ELISA

The intestinal samples were weighed (0.1 g) and then cut to pieces. Cut tissue and 900 μL PBS buffer were added to a glass homogenizer and ground on ice. To further lyse tissue cells, the homogenate can be ultrasonically crushed or repeatedly thawed. Finally, the homogenate was centrifuged in an ice centrifuge at 5000 rpm for 5–10 min, and the supernatant was taken and sub-packaged for later use. The concentrations of IgA, IgD, IgM, IgG, IL-1β, IL-2, IL-6, and IL-8 were analyzed by ELISA method. The test kits were all purchased from Quanzhou Kenuodi Biology Co., Ltd. The specific test methods were all carried out according to the kit instructions.

### 5.5. Intestinal Morphology

The samples of intestinal tissue were soaked in 4% paraformaldehyde for 4 h, and then transferred to 70% ethanol. These samples, which were placed in a processing box, were dehydrated with gradually increasing concentration of ethanol gradients and embedded in paraffin blocks. The tissue sections were dewaxed using xylene, rehydrated with a series of decreasing alcohol gradients, and then rinsed by PBS. After staining by hematoxylin and eosin (HE), the tissue sections were dehydrated by ethanol and xylene. The villi height (VH) and crypt depth (CD) of intestinal samples were observed and measured under the microscope [45].

### 5.6. Goblet Cells

The tissue sections of intestinal samples were stained using combined Alcian Blue and Periodic Acid Schiff (AB-PAS), after which the goblet cells were observed under 400× microscope [46]. Five slices were selected from each intestinal segment and five villi were selected in each field. The 1500-μm villi were observed and the number of goblet cells at the top of villi was counted.

### 5.7. Immunohistochemistry (IHC) of Intestinal Samples

Paraffin sections (5-μm-thick) were pasted on the glass slides treated with poly-lysine. After dewaxing and debenzened, the thermal remediation of antigen was carried out in sodium citrate buffer (pH 6.0, 0.01 M). The sections were incubated using 3% H_2_O_2_, which were used to cut off the activity of endogenous peroxidase, followed by 10% calf serum. In this experiment, the primary antibody was polyclonal rabbit antibodies against IL-1β, IL-2, IL-6, and IL-8 (1:100, BIOSS, Beijing, China) and the secondary antibody was the Polink-2 plus immunohistochemical assay kit (PV-9001, ZSBIO, Beijing, China). Immunostaining was detected using DAB kit (Tiangen, Beijing, China). After the sections were hematoxylin redyed, dehydrated, hyalinized, and sealed in clear resin, the immune positive cells (brown-yellow) were observed under a microscope. The image analysis software (Image Pro-Plus 6.0, Media Cybernetics, MD, U.S.A.) was used for evaluate the amount of cell staining. Total cross-sectional integrated optical density (IOD) and the light density of the single intestinal villi (SIOD) were calculated, which was used for the quantitative analysis of positive cells in the duodenum, jejunum, and ileum of different groups.

### 5.8. Western Blot

The total protein were separated and extracted from the small intestine tissues using the Protein Extraction Kit (Beyotime, Shanghai, China). For western blot, primary antibodies, specific antibodies against polyclonal rabbit antibody IL-1β, IL-2, IL-6, and IL-8 (1:2000, BIOSS, Beijing, China), or monoclonal mouse β-actin (1:300, Beyotime, Shanghai, China), were incubated at 4 °C for 12 h. The secondary antibody, goat anti-rabbit IgG, or goat anti-mouse IgG, were labeled by HRP (1:2000, Beijing, China), was incubated for 2 h. The bands were colored using BeyoECL Plus P0081 kit (Beyotime, Shanghai, China) and taken a picture by FusionCapt Advance FX7 (Fusion FX, OSTC), and then quantitative data of bands were obtained using Image software (image Pro-Plus 6.0, Media Cybernetics, Silver Spring, USA).

### 5.9. qRT-PCR

Total RNA of intestinal samples were isolated using Trizol Reagent kit (Invitrogen Co., Carlsbad, CA, USA). After the quality and concentration were detected by spectrophotometer, the total RNA was reverse-transcribed immediately. The procedure of reverse transcription was in accordance with the instructions of PrimeScript^@RT^ Master Mix Perfect Real Time kit. Primer sequence of the IL-1β, IL-2, IL-6, IL-8, and GAPDH are listed in Table 4, and all the genes’ sequences were quoted from NCBI. A total volume of 20 μL of reaction system included 10 μL SYBR Promerx Ex Taq, 0.4 μL upstream primers (10 μM/L), 0.4 μL downstream primers (10 μM/L), 0.4 μL ROX Reference Dye, 2 μL cDNA, and 6.8 μL dH_2_O. The reactions of real-time PCR were carried out at 95 °C for 15 s, followed by 40 cycles at 95 °C for 15 s, and 60 °C for 60 s. The relative expression of mRNA was calculated as being equal to 2^−ΔΔCT^ [47].

### 5.10. Statistics

The software of SPSS 21.0 for Windows was used for all statistical analysis in this study. One-way ANOVA was used to analyze all data, including the data from image scanning. And the mean values were compared between individual groups using the Tukey’s test. All results were represented in terms of mean ± standard deviation, and *p* < 0.05 was implied a significant difference.

## Figures and Tables

**Figure 1 toxins-11-00473-f001:**
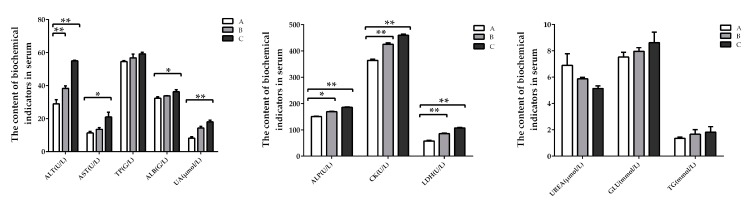
Effect of deoxynivalenol (DON) on serum biochemistry of weaning rabbits (Mean ± SD, n = 10). TP, total protein; ALB, albumin; UREA, blood urea; UA, uric acid; AST, glutamic oxalacetic transaminase; ALT, alanine amiotransferase; ALP, alkaline phosphatase; CK, creatine phosphokinase; GLU, glucose; TG, triglyceride; LDH, lactic dehydrogenase. Group A, control group; groups B and C, DON was added at 0.5 mg/kg BW and 1.5 mg/kg BW, respectively. * and ** indicate the significant difference at *p* < 0.05 and *p* < 0.01, respectively.

**Figure 2 toxins-11-00473-f002:**
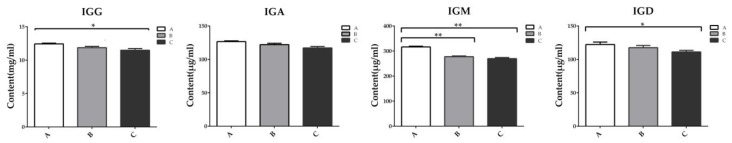
The concentration of immunoglobulin in serum induced by DON (Mean ± SD, n = 10). Group A, control group, groups B and C, DON was added at 0.5 mg/kg BW and 1.5 mg/kg BW, respectively. * and ** indicate the significant difference at *p* < 0.05 and *p* < 0.01, respectively.

**Figure 3 toxins-11-00473-f003:**
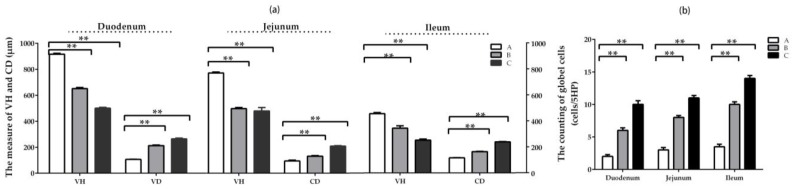
The intestinal histomorphology in different intestinal segments induced by DON (Mean ± SD, n = 5). (**a**) The measure of height of villi (VH) and depth of crypt (CD). (**b**) The counting of goblet cells. Group A refers to control group, while groups B and C indicate groups where DON was added at 0.5 mg/kg BW and 1.5 mg/kg BW, respectively. Statistically significant difference: * at *p* < 0.05 and ** *p* < 0.01.

**Figure 4 toxins-11-00473-f004:**
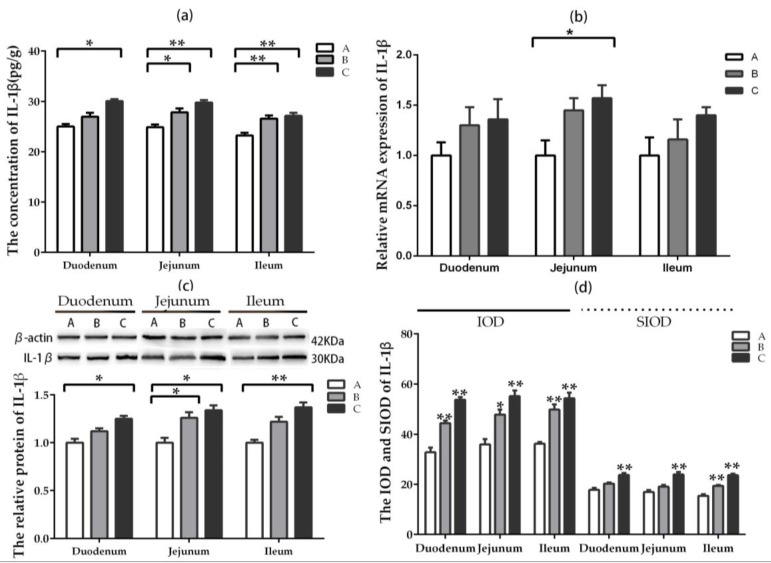
Expression and distribution of IL-1β in different intestinal segments (Mean ± SD, n = 5). (**a**) The concentration of IL-1β examined by ELISA. (**b**) The relative mRNA expression of IL-1β detected by qRT-PCR. (**c**) The relative protein expression of IL-1β analyzed by western blot. (**d**) The IOD and SIOD value of IL-1β examined by IHC method. Group A means the control group, while groups B and C refer to groups to which DON was added at 0.5 mg/kg BW and 1.5 mg/kg BW, respectively. * and ** indicate the significant difference at *p* < 0.05 and *p* < 0.01, respectively.

**Figure 5 toxins-11-00473-f005:**
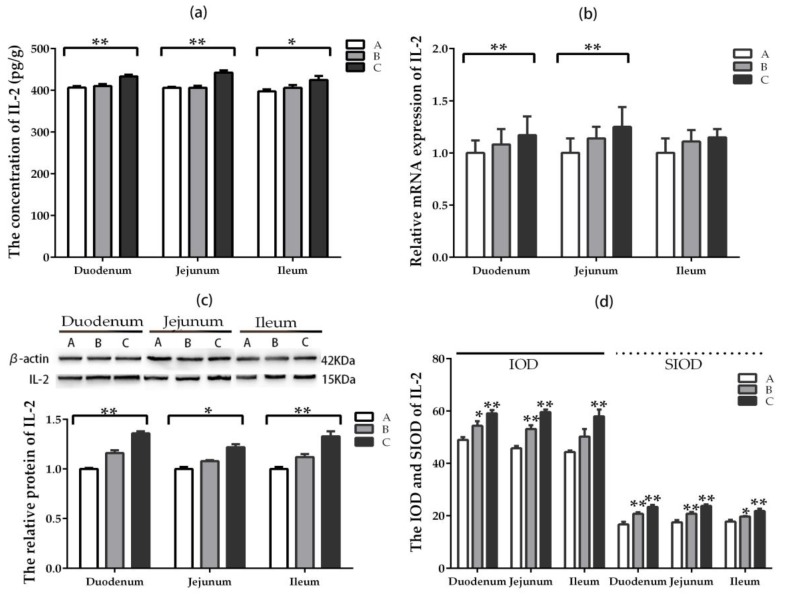
Expression and distribution of IL-2 in different intestinal segments (Mean ± SD, n = 5). (**a**) The concentration of IL-2 examined by ELISA. (**b**) The relative mRNA expression of IL-2 detected by qRT-PCR. (**c**) The relative protein expression of IL-2 analyzed by western blot. (**d**) The IOD and SIOD value of IL-2 examined by IHC method. Group A means the control group, while groups B and C refer to groups where DON was added at 0.5 mg/kg BW and 1.5 mg/kg BW, respectively. * and ** indicate the significant difference at *p* < 0.05 and *p* < 0.01, respectively.

**Figure 6 toxins-11-00473-f006:**
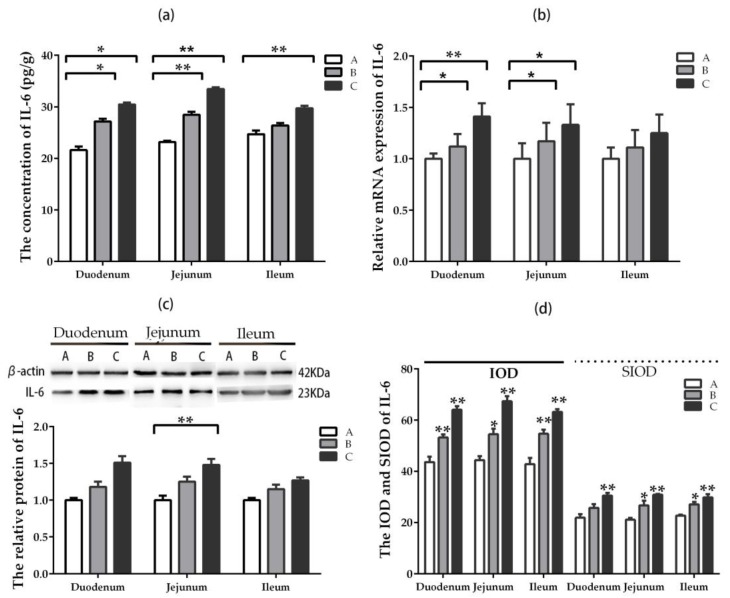
Expression and distribution of IL-6 in different intestinal segments (Mean ± SD, n = 5). (**a**) The concentration of IL-6 examined by ELISA. (**b**) The relative mRNA expression of IL-6 detected by qRT-PCR. (**c**) The relative protein expression of IL-6 analyzed by western blot. (**d**) The IOD and SIOD value of IL-6 examined by IHC method. Group A means the control group, while groups B and C refer to groups where DON was added at 0.5 mg/kg BW and 1.5 mg/kg BW, respectively. * and ** indicate the significant difference at *p* < 0.05 and *p* < 0.01, respectively.

**Figure 7 toxins-11-00473-f007:**
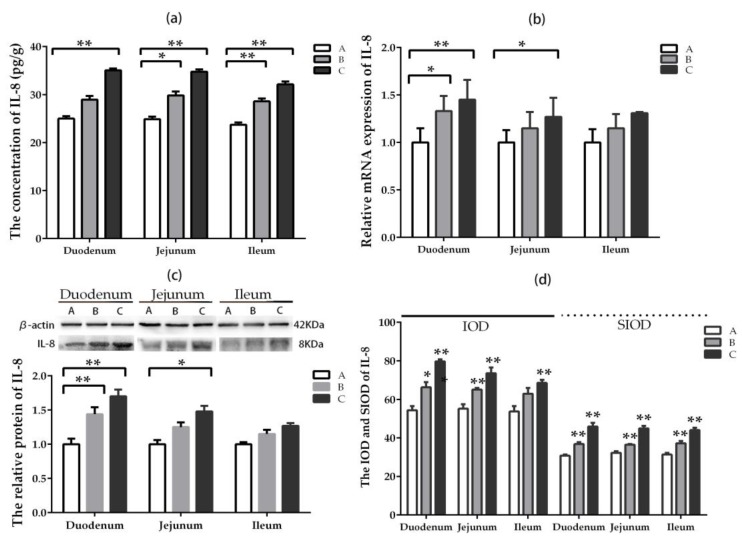
Expression and distribution of IL-8 in different intestinal segments (Mean ± SD, n = 5). (**a**) The concentration of IL-8 examined by ELISA. (**b**) The relative mRNA expression of IL-8 detected by qRT-PCR. (**c**) The relative protein expression of IL-8 analyzed by western blot. (**d**) The IOD and SIOD value of IL-8 examined by IHC method. Group A means the control group, while groups B and C refers to groups where DON was added at 0.5 mg/kg BW and 1.5 mg/kg BW, respectively. * and ** indicate the significant difference at *p* < 0.05 and *p* < 0.01, respectively.

**Table 1 toxins-11-00473-t001:** Growth performance of weaning rabbits induced by DON (Mean ± SD, n = 15).

Group	Initial Weight (g)	Final Weight (g)	ADG (g)	ADFI (g)	F/G
A	1079 ± 35	1634 ± 61	23.13 ± 1.78 ^a^	133 ± 9	5.72 ± 0.43 ^b^
B	1099 ± 43	1644 ± 65	22.73 ± 1.09 ^a^	129 ± 9	5.66 ± 0.31 ^b^
C	1103 ± 46	1621 ± 59	21.55 ± 1.30 ^b^	126 ± 8	5.86 ± 0.49 ^a^

The different letters in a column indicate the significant difference (*p* < 0.05). Group A, control group; groups B and C, DON was added at 0.5 mg/kg BW and 1.5 mg/kg BW, respectively.

**Table 2 toxins-11-00473-t002:** Relative organ weight (g/kg) ^1^ of weaning rabbits induced by DON (g/kg, Mean ± SD, n = 5).

Group	Liver	Kidney	Spleen	Stomach
A	35.77 ± 1.12 ^a^	8.00 ± 0.23	0.70 ± 0.08 ^a^	16.73 ± 1.04
B	29.07 ± 0.57 ^b^	7.17 ± 0.26	0.54 ± 0.04 ^b^	16.59 ± 0.27
C	30.87 ± 1.45 ^b^	7.84 ± 0.33	0.50 ± 0.05 ^b^	17.45 ± 0.80

The different letters in a column indicate the significant difference (*p* < 0.05). ^1^ Relative organ weight (g/kg) is equal to organ weight (g) divided by final weight of rabbits (kg). Group A, control group; groups B and C, DON was added at 0.5 mg/kg BW and 1.5 mg/kg BW, respectively.

**Table 3 toxins-11-00473-t003:** The ingredient composition of the basic diets (as feed basis).

Ingredient (%)		Calculated Composition	
Maize	14	Dry matter	88.64
Soybean meal	17	Crude protein	20.05
Wheat bran	13	Crude fiber	18.78
Corn germ meal	19	Crude ash	10.45
Rice hulls	10	Crude fat	3.34
Soybean straw powder	7	Calcium	0.72
Alfalfa	10	Total Phosphorus	0.55
Malt Sprout	5	Digestible energy (MJ/kg)	10.06
Sweet wormwood	3.5		
Premix material ^1^	1.5		
Total	100		
**Content of Mycotoxin (μg/kg) ^2^**			
Deoxynivalenol (DON)	23.18		
Zearalenone (ZEA)	257.76		
Aflatoxin B1 (AFB1)	7.08		

^1^ Premix material provided per kg feed: VA 12000 IU; VB1 1.50 mg;VB2 5 mg; VB3 4 mg; VB5 50 mg; VB6 0.5 mg; VB11 2.5 mg; VB12 0.02 mg; choline 600 mg; biotin0.2 mg; VD 3100 IU; VE 50 mg; VK 31.5 mg; Fe 60 mg; Zn 60 mg; Cu 40 mg; Mn 9 mg; Se 0.2 mg; mountain flour 15,000 mg; NaCl 5000 mg; lysine 1500 mg; methionine 1000 mg. ^2^ Measured by LC-MS/MS method.

**Table 4 toxins-11-00473-t004:** Primer sequence used in qRT-PCR.

Target Gene	GenBank Number	Primer Sequence (5′-3′)
IL1β-F	*NC_013670*	TTCCGGATGTATCTCGAGCA
IL1β-R		GTGGATCGTGGTCGTCTTCA
IL2-F	*NC_013683*	GCTTCGATGCCAGTGCATAA
IL2-R		CAGGCAGAGTTCTCTTCCATCA
IL6-F	*NC_013678*	GCCAACCCTACAACAAGA
IL6-R		AGAGCCACAACGACTGAC
IL8-F	*NC_013683*	TGCCCAAGAAGGTCACAGAA
IL8-R		ACTCGATGCTGAGATGATGCT
GAPDH-F	*NM_001082253*	TGCCACCCACTCCTCTACCTTCG
GAPDH-R		CCGGTGGTTTGAGGGCTCTTACT

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
