# Peer review of "The Effect of Low and High Dose Deoxynivalenol on Intestinal Morphology, Distribution, and Expression of Inflammatory Cytokines of Weaning Rabbits"

_toxins, 2019, doi:10.3390/toxins11080473_

Round 1

Reviewer 1 Report

The manuscript is well written and well conducted. I only have some minor remarks:

Line 239. Replace , with . (Before In particular)

Line 273. Replace "the" with "The". 

Line 296. Replace "ra" with the proper word. 

A better proportion for the Figure 5 (b).

Make a stronger conclusion. Why this study is so important to be published, how these results cam help future studies, what would be the next step in the investigation?

Reviewer 2 Report

General comment

Interesting study for which I have 3 major criticisms:

1- lack of information on the dose at which rabbits were exposed in mg/kg body weight (I only found the level of DON in water but water consumption was not given)

2- no data on health, body weight and organs weight in the manuscript

3- no semi-quantitative analysis of western blot and immunohistochemistry

I think all these points are relatively easy to correct and that correction would strongly increase the overall merit/interest of the study.

Specific comments

All the manuscript, the word « treatment » should be changed to « group »

L9: I do not understand the dose. I suppose DON was diluted in water, so 0.5mg/L and 1.5 mg/L? Because this exposure is not common you should also calculate an exposure in mg/kg BW

L10: every day ok but how many days?

L13 and all the manuscript: check for typing errors (space interval between characters)

L28: I understand what you wanted to say but DON is not “chemically stable”, as it is easily destroyed by ozone and is not orally stable in ruminants

L32-55: clarify the species

L61: because of the way of exposure I think you have to make a paragraph on the consequences the way of exposure has on toxicity with at least to points:

- the real exposure of each animal (in link with consumption and risk of consumption refusal )

- the bioavailability of the toxin (which can vary with the medium in which it was adminsitered)

L68: add a paragraph on clinical signs, body weight… what was observed during the study. Also, add drink intake and calculation of animal exposure in mg/kg BW

L101: “There were no significant differences” two times

Figure 4 can be submitted as supplementary file as the photographs in figure 5 to 9. As they are submitted these photographs are difficult to read (weak size/resolution) and have weak scientist interest because semi-quantitative analysis was done/could be done. Submitting these photographs as supplementary data would permit to have picture of large size

L134-221, Figures 6 to 9: You have measured the effect of DON using 3 methods: ELISA, Western blotting, immunohistochemistry. This is very interesting but in my opinion the way the results are presented is not the best choice as quantitative analysis is only done for ELISA. Why no quantitative analysis was done for western-blot and immuhistochemistry? This would permit to make statistical analysis of the effect of DON.

L223: first discuss the exposure

L227-230: add bioavailability and between species variation

L239: add dose/level/duration at which those effects are observed in other species

L246: taking into account previous remarks add information about the dose at which these effects were observed in rabbits. Accordingly this will permit to known whether rabbits are highly sensitive as pigs or rather resistant.

L260-260: same comment, it is difficult to discuss “sensitivity” in the absence on dose

L281-288: as said before, a semi-quantitative analysis would strongly increase the impact of the discussion of the effects and the dose-response

L327: if animals were “individually housed in metabolism cages” that means data on drink and feed intake are probably available. Also, body weigh should be available.

L330-332: should be added to the abstract

L341: there was no postmortem examination, body and tissue weight? Also, as said before, drink intake and calculation of DON intake is absolutely necessary

L358: “Intestinal”

L371: clarify number of sample/slide per animal

L389: why no quantitation of the number of positive cells?

L400: image analysis should include the measure of blot density and comparison with actin for each sample. Then analysis of the ratio can be done and results can be compared with statistics

Round 2

Reviewer 2 Report

Thank you for the change made, I just have some minor comments/suggestions to complete:

L9 and all the MS when necessary : I suggest to change “mg/kg.BW/d” to “mg/kg BW/d”

L11: please add a short sentence to clarify these effects were observed at a DON level that has an effect on heath and performances (decrease in BWG, FC ratio)

Table 1: add a foot not explaining abbreviations used.
I suggest to remove not significant number after dot : change “1078.63±35.14” to “1079±35”, “23.13±1.78” to “23.13±1.78” and keep only “5.72±0.43” with two digits. This will make easy the reading of the Table

L214: please avoid the use of “ppm” in the discussion and change to something like: “usually monitored at the mg/kg feed, also known as the “ppm level”.”

L218-221 and L234-239: It would be easier to have exposure in mg/kg feed. Data can be recalculated from publications when not available, an estimation is sufficient. If you prefer to keep mg/kg BW (may be easier for you), you have to clarify exposure in L222-231 in mg/kg BW

L263: change “higher dose” to the dose used

L293: please add a short sentence to clarify these effects were observed at a DON level that has an effect on heath and performances (decrease in BWG, FC ratio)

Author Response

Responds to the reviewer’s comments:
L9 and all the MS when necessary : I suggest to change “mg/kg.BW/d” to “mg/kg
BW/d”
--- Great thanks for your professional suggestion. all “mg/kg.BW/d” has been
change to “mg/kg BW/d” in revised manuscript.
L11: please add a short sentence to clarify these effects were observed at a DON
level that has an effect on heath and performances (decrease in BWG, FC ratio)
--- Great thanks for your professional suggestion. the sentence about the
performance have been added in the revised manuscript.
Table 1: add a foot not explaining abbreviations used.
I suggest to remove not significant number after dot : change “1078.63±35.14” to
“1079±35”, “23.13±1.78” to “23.13±1.78” and keep only “5.72±0.43” with two digits.
This will make easy the reading of the Table
--- Great thanks for your professional suggestion. the dot after number have been
adjusted in revised manuscript.
L214: please avoid the use of “ppm” in the discussion and change to something
like:
--- Great thanks for your professional suggestion. “ often at the ppm level” has
been change to “usually monitored at the mg/kg feed.”
L218-221 and L234-239: It would be easier to have exposure in mg/kg feed. Data
can be recalculated from publications when not available, an estimation is sufficient.
If you prefer to keep mg/kg BW (may be easier for you), you have to clarify exposure
in L222-231 in mg/kg BW.
--- Great thanks for your professional suggestion. All data about DON dose at
mg/kg BW was the original published data, and all dose of DON have been
calculated to mg/kg BW in revised manuscript.
L263: change “higher dose” to the dose used
--- Great thanks for your professional suggestion. the dose used was added in the
revised manuscript.
L293: please add a short sentence to clarify these effects were observed at a DON
level that has an effect on heath and performances (decrease in BWG, FC ratio)
--- Great thanks for your professional suggestion. the conclusion has been
changed to “ the addition of DON can induced the lower feed efficiency and weight

loss, as well as damaged the integrity of intestine segment, changed the distribution
and expression of inflammatory cytokines.” In the revised manuscript.
As you suggested, we have submitted the revised version .If there is any points
need to be clarified, please tell me by Email.
Great thanks for your special work for this manuscript.
